# Effects of caffeine consumption combined with listening to music during warm-up on taekwondo physical performance, perceived exertion and psychological aspects

Slaheddine Delleli[1,2], Ibrahim Ouergui[3,4]*, Hamdi Messaoudi[1,2], Christopher Garrett Ballmann[5], Luca Paolo Ardigò[6‡]*, Hamdi Chtourou[1,2‡]

1 Research Unit, Physical Activity, Sport and Health, UR18JS01, National Observatory of Sport, Tunis, Tunisia, 2 High Institute of Sport and Physical Education of Sfax, University of Sfax, Sfax, Tunisia, 3 High Institute of Sport and Physical Education of Kef, University of Jendouba, Kef, Tunisia, 4 Research Unit, Sports Science, Health and Movement, University of Jendouba, El Kef, Tunisia, 5 Department of Human Studies, University of Alabama at Birmingham, Birmingham, Alabama, United States of America, 6 Department of Teacher Education, NLA University College, Oslo, Norway

‡ LPA and HC share last authorship on this work.
* ouergui.brahim@yahoo.fr (IO); luca.ardigo@nla.no (LPA)

**Data Availability Statement:** All relevant data are within the manuscript and its Supporting information file.

## Abstract

The effects of caffeine (CAF) and music have been well documented when used separately, but their combined effects are not yet studied. Thus, the present study assessed the acute effects of combining a low dose of CAF with listening to music during warm-up on taekwondo physical performance, perceived exertion (RPE), and psychological responses during taekwondo-specific tasks in male elite athletes. In a double-blinded, randomized, placebo-controlled crossover study design, male taekwondo athletes (n = 16; age: 18.25 ± 0.75 years) performed the taekwondo-specific agility test (TSAT), 10 s frequency speed of kick test (FSKT-10s) and the multiple version of FSKT (FSKT-mult) under the following conditions: 1) CAF without music (CAF+NoM), 2) placebo (PL) without music (PL+NoM), 3) CAF with music (CAF+M), 4) PL with music (PL+M), 5) no supplement with music (NoS+M) and no supplement without music (control). RPE, feeling scale (FS), felt arousal scale (FAS) and physical enjoyment (PACES) were determined after each test. Findings showed the CAF+M condition induced better performances than other conditions for TSAT, FSKT-10s, FSKT-mult, RPE, FAS and FS and PACES post FSKT-10s (all p<0.05). Moreover, CAF+M resulted in better responses than other conditions for PACES post TSAT (p<0.05) with the exception of CAF+NoM. Likewise, CAF+M condition induced better physical enjoyment than PL+NoM, NoS+M and PL+M conditions post FSKT-mult (p<0.05). Combining low dose of CAF with music during warm-up was an effective strategy that induced greater effects than their isolated use during taekwondo specific tasks.

**Funding:** The authors received no specific funding for this work.

**Competing interests:** The authors have declared that no competing interests exist.

## Introduction

In order to enhance athletic performance or reduce fatigue-generated symptoms, innumerable ergogenic resources have been proposed. In particular, sports supplements use is a quite prevalent strategy among athletes [1]. As a psychoactive substance, caffeine (CAF) (1,3,7-trimethyl-xanthine) is among the most commonly consumed supplements due to its ergogenic effects [2]. Ergogenic potential of CAF intake has been widely established for many exercises types, including muscular endurance/ strength, anaerobic power, and aerobic endurance [3]. The ergogenic potential of CAF is generally explained by its capacity to block the adenosine receptors $A_1$, $A_{2A}$, and $A_{2B}$, due to their similarity in term of chemical structures [4]. In addition, CAF supplementation has been shown to promote a positive mood [5, 6], to increase alertness and reduces the feeling of fatigue [7].

In combat sports, CAF has been reported to improve different performance aspects involving isometric strength, anaerobic power, reaction time, and anaerobic metabolism [8]. However, there were inconsistent findings about its ergogenic potential which was mainly explained by the dose consumed, form, timing, inter-individual variability, and genetic background as well as exercise modality [8]. Specifically, taekwondo is a striking combat sports where previous studies reported beneficial effects of CAF consumption on specific performance [9, 10]. In fact, using different testing procedures (i.e., tests with different physical ability and duration) and dosing protocols (3–6 mg/kg of body mass), CAF improved agility, reaction time, offensive actions, metabolism and psychological responses to exercise [10–12].

Although the ergogenic effects of CAF when consumed alone has been well documented, its combination with other supplements (e.g., sodium-bicarbonate, carbohydrate) [13, 14] or other training strategy (e.g., conditioning activity during warm-up) [12] have resulted in synergistic effects. However, since the psychological state is a key determinant of athletic performance [15], using a psycho-affective stimulant could be an effective strategy to enhance performance. In this consideration, music has been well presented as an ergogenic aid with psychological effects [16]. In fact, music' effects on exercise performance are multifaceted, with potential benefits in a variety of exercise modalities and athletic populations [17]. These benefits were linked to the music ability to promote dissociation from exertion, increasing motivation, activity enjoyment, self-confidence, feelings of power, and regulated arousal [17–20]. The ergogenic effect of music has been well reported specially when it was played in the context of preference [18]. However, listening to music while competing may not be feasible for many competitors [21] such the case of combat sports athletes. Therefore, listening to music during warm-up may allow for sustained motivation throughout repeated efforts during competition [22, 23].

Although moderate to high doses of CAF have been well reported to have ergogenic effects, its use may be accompanied with some adverse effects (e.g., tachycardia, anxiety, headache, abdominal discomfort and insomnia) [24, 25]. Consequently, it is difficult to consume such high CAF doses through a regular diet [26]. However, the combined effects of a low dose of CAF with other stimuli could be of similar or greater benefits than its isolated use. In this context, Ouergui et al. [12] reported that 3 mg·kg$^{-1}$ of CAF consumption combined with the use of a plyometric conditioning activity resulted in greater physical performances and psychological aspects than the CAF consumed alone, without reported side effects. To the best of our actual knowledge, while the effects of CAF and music, both in single-use, are well established, no study has investigated the combined effects of a low dose of CAF ingestion and listening to music during warm-up on physical performances and psychological responses. Therefore, the present study aims to investigate the effects of combining CAF with preferred warm-up music on taekwondo-specific agility and kicking performances and the associated psychological

responses. While music is a psychological ergogenic aid that serves to improve exercise behaviors and affective states [16], CAF is a pharmaceutical substance which showed its effectiveness to enhance neurological and physiological responses to exercise [27]. Given that performance is an integration of physical capacities, physiological responses and psychological behaviors [28], it was hypothesized that combining a low dose of CAF with listening to music during warm-up would result in greater physical performances and psychological responses.

## Materials and methods

### Participants

The required sample size was calculated by the G*Power software (Version 3.1.9.4, University of Kiel, Kiel, Germany) using the F test family (repeated measures, within factors), with six conditions. The power analysis revealed that a total sample size of 14 would be sufficient to find medium significant effects of condition (effect size f = 0.294, $\alpha$ = 0.05) with an actual power of 82%. To avoid the lack of statistical power which may be induced by the risk of dropout of some participants from the study, 16 male taekwondo athletes from the national team (Mean ± SD; age: 18.25 ± 0.75 years; body mass: 60.92 ± 8.96 kg; height: 182 ± 6.84 cm) volunteered to participate in the present study. Athletes were recruited following a convenience sampling based on the following eligibility criteria: a) being an elite athlete with at least 8 years of taekwondo experience; b) do not suffer from any restrictions to sports practice or hearing impairments; c) have at least 17 years old, and d) being non-smoker. Based on the questionnaire of Bühler et al. [29], athletes were considered as low CAF consumers (i.e., mean habitual CAF consumption = 1.33 ± 0.42 mg/kg). The participants were asked to follow the same diet, avoid alcoholic substances and vigorous exercise, and restrain from CAF consumption (in drinks and supplements) 48h before each experimental session. All participants were informed about the procedures, the possible risks and discomforts involved in the investigation and they and/or their parents (only for 2 athletes) signed a written informed consent form. This study was conducted in accordance with the last Declaration of Helsinki and the protocol was fully approved by the University of Jendouba Research Ethics Committee (CPP SUD N˚ 0332/2021) before data collection. The study was conducted during the period from the 1st of August to the 18th of September 2022 (1 week for music selection and familiarization + 6 sessions, with 7 days in between). The participants' flow diagram is presented in Fig 1.

### Experimental design

This is a double-blind, counterbalanced, crossover study design aiming to investigate the acute effects of CAF supplementation combined with listening to music during warm-up on the subsequent 10 s frequency speed of kick test (FSKT-10s), multiple frequency speed of kick test (FSKT-mult), and taekwondo specific agility test (TSAT) performances, the felt arousal scale (FAS), the physical activity enjoyment scale (PACES), the feeling scale (FS) and the rating of perceived exertion (RPE) in elite male taekwondo athletes. The study was conducted following the Consolidated Standards of Reporting Trials (CONSORT) guidelines for randomized crossover trial [30] (S1 Table).

During the first visit, athletes were familiarized with the testing procedure and anthropometrics characteristics measurements were conducted. During the testing sessions, athletes were submitted to six conditions (five experimental conditions and one control) in a randomized counterbalanced order (i.e., the 16 athletes were split into two groups of two and four groups of three). Specifically, in a double-blind fashion, athletes ingested 3 mg·kg$^{-1}$ of body mass of CAF or placebo (PL) -for all-purpose bleached flour. The CAF dose was chosen due to its safety as well as its effectiveness to improve performance in combat sports was previously

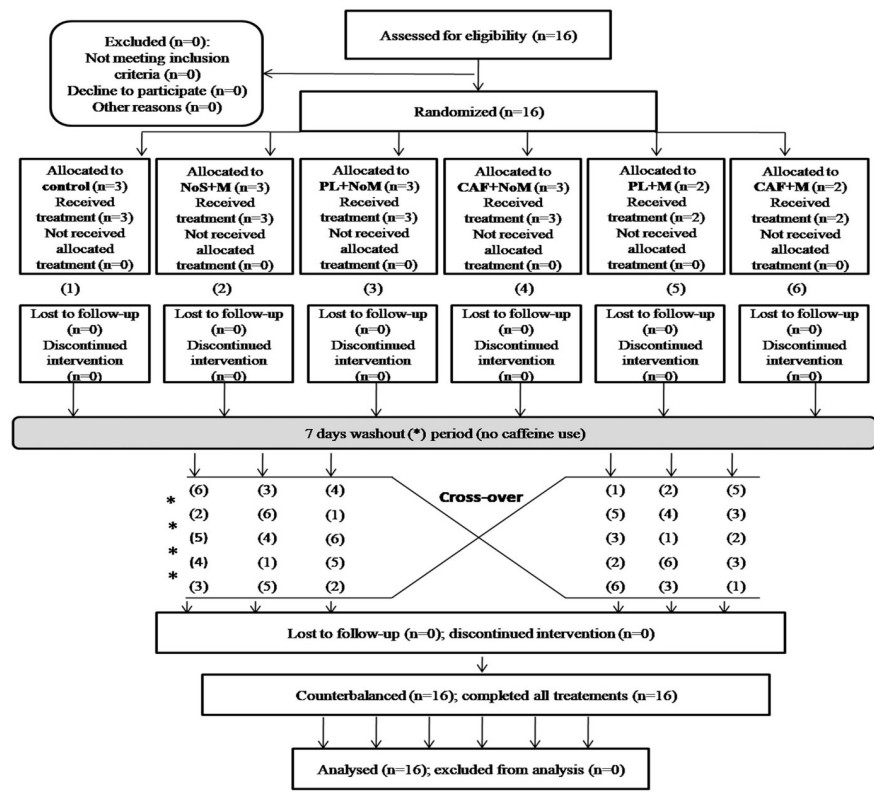

**Fig 1. Participants flow diagram.** PL: placebo, CAF: caffeine, M: music, NoS: no supplement, NoM: no music, *: washout period.

reported [12, 31]. Fifty min after supplementation, athletes performed two warm-up protocols, consisting of a standard warm-up (i.e., 8 min of running at 9km/hr and taekwondo-specific techniques execution) without music (control), or performing warm-up while listening to a priori selected music (i.e., 8 min warm-up with music), both followed by two min of rest. Thereafter, athletes performed the specific tests through six different conditions: 1) no supplement and no music (control), 2) no supplement with music (NoS+M), 3) placebo and no music (PL+NoM), 4) CAF and no music (CAF+NoM), 5) placebo with music (PL+M), and 6) CAF with music (CAF+M). In each testing session, arousal after warm-up was measured using the felt arousal scale (FAS). Moreover, after each test, athletes rated their FS, FAS, PACES, and RPE. To control the possible side effects of CAF supplementation, a gastro-intestinal discomfort questionnaire was administered before supplementation and after each session was adapted from previous study [32]. The sessions were separated by an interval of seven days to allow sufficient recovery between sessions and to ensure CAF washout [33]. All sessions were conducted at the same time of day (17:00–18:00 p.m) to overcome the diurnal variation of the performance.

## Supplementation procedure

For the supplement conditions, each athlete received a 3 mg·kg$^{-1}$ of CAF or PL in a crossover regimen. Both CAF and PL were administered after being dissolved in 200 ml of water. The dose of CAF was chosen as it has shown its effectiveness to improve male taekwondo performance while minimal to no side effects in this population were reported [12]. On testing

days, the supplements were taken 60 min before performing the taekwondo tests. The preparations were administered to each participant in containers marked with a unique code. To ensure the double-blind procedure, the preparations were made in advance by a research team member who did not directly participate in the investigation. Participants were instructed to not discuss or compare tastes and the investigators along with athletes' coaches supervised consumption compliance. To check the success of blinding, each subject was asked to identify what the treatments after consumption took. Verbal questioning after every session indicated that almost all participants were unable to distinguish between the supplements.

## Music selection procedure

Music selection was performed before familiarization and testing sessions according to procedures adopted from previous investigations [17, 18, 34] on music preference and exercise performance. Each athlete was asked to select his favorite music and rated it using the Brunel Music Rating Inventory-2 [35] to control the motivational quotient of the selected music. The selected music was accepted to be used during music conditions as long as it had a tempo $\geq$ 120 bpm to ensure that the music was stimulatory in nature [36]. Mean tempo of the selected music from all participants was 142 ± 18.6 bpm. Music was listened to using headphones connected to the athlete's personal mobile phone. Single truck for each athlete was used for all sessions and the music volume was standardized at the same level (i.e., 80 db) for all participants. The decibel level of the music was administered under check using the application Decibel: dB sound level meter (developer Vlad Polyanskiy). For the sessions using music, the preferred track for each participant was played until the completion of the warm-up session. In case it ended before the warm-up completion, music was looped. For NoM conditions, participant wore their headphones while warming-up with no music was played to assure the same testing conditioning.

## Testing procedures

**Physical performance** *Taekwondo specific agility test*. During the test, athlete began from a guard position with both feet behind the start/finish line. At his discretion, athlete moved as quickly as possible towards the center point. Then, following his own preference, he moved in sideways towards partner 1 and performed a roundhouse kick with the lead leg. Subsequently, he turned and shifted to partner 2 and performed a right another roundhouse kick with the other lead leg. Next, he returned to the center moved forward to partner 3 in a guard position and performed a double roundhouse kick. Finally, the athlete returned to the start/finish line [37]. The completion time was measured using photocells (Brower Timing Systems, Salt Lake City, UT, USA). Each athlete performed three trials and the best one was used for analysis. The intra-class correlation coefficient (ICC) for test-retest in the present study was 0.88.

*Ten seconds frequency speed of kick test*. During the test, the athlete must perform the maximum number of bandal-chagui against a punching bag by alternating the right and left leg [38]. The number of techniques performed during the 10s of test represented the performance index [38]. The ICC for the test-retest in the present study was 0.83.

*Multiple frequency speed of kick test*. The FSKT-mult presented the multiple version of the FSKT-10s. In fact, athlete performed five sets of FSKT-10s with a 10 s rest interval between repetitions. Performance was determined by the total number of kicks in 5 sets. The ICC for the test-retest in the present study was 0.77.

## Perceived exertion and psychological measures

**Rating of perceived exertion.** Perceived exertion was assessed using the CR-10 Borg scale [39]. This is a scale ranging from "0" to "10", with corresponding verbal expressions, which gradually increases with the intensity of perceived sensation (0 = nothing at all; 1 = Very week; 2 = week; 3–4 = Moderate; 5–6 = strong; 7–9 = very strong; and 10 = extremely strong). Over sessions, athletes were asked to rate their RPE just after each test.

**The physical activity enjoyment scale.** PACES was used to detect the level of pleasure and enjoyment of participants [40]. The original version of this scale with 18 items was used. Items involved 11 negative and 7 positive items measured through a 7-point score ranging from 1 to 7 [40]. Based on the sum of total responses for each athlete, the score could range from 18 to 126.

## Feeling scale

The affective responses were assessed after each test using the feeling scale [41]. The FS utilizes a single-item 11-point bipolar rating scale ranging from -5 to +5, with the stem "How do you currently feel?". Anchors are given at 0 (Neutral) and all odd integers, ranging from "Very Bad" at -5 to "Very Good" at +5.

## Felt arousal scale

The Felt arousal scale was used to measure arousal along a 6 point scale ranging from low arousal (1 point) to high arousal (6 points) [42]. The participants were instructed to mark the scale on the basis of their perceived activation after each test.

## Gastro-intestinal discomfort questionnaire

This questionnaire was used to check the presence of any gastro-intestinal problems before and after supplementation. The questionnaire included gastro-intestinal symptoms that could be associated with CAF intake (i.e., nausea, vomiting, headache, heartburn, abdominal pain, diarrhea, breathlessness and constipation) [24, 25]. Each symptom was considered: 1) Absent, 2) mild, 3) moderate, or 4) severe. As a tool for the screening of side effects, this questionnaire should not require large simple size to be effective [32].

## Statistical analyses

The statistical analysis was performed using SPSS 20.0 statistical software (IBM corps., Armonk, NY, USA). Data were presented as mean and standard deviation and Median and Interquartile range values were reported for non-normal distribution data. The Shapiro-Wilk test was used to check and confirm the normality of data sets, and the Levene test was used to verify the homogeneity of variances. Sphericity was tested using the Mauchly test. For TSAT, a one-way analysis of variance (ANOVA) (condition) with repeated measurements was used, with Bonferroni was used as post hoc test. Standardized effect size analysis (Cohen's d) was used to interpret the magnitude of differences between variables and considered as: trivial ($\leq 0.20$); small ($\leq 0.60$); moderate ($\leq 1.20$); large ($\leq 2.0$); very large ($\leq 4.0$) (very large); and extremely large ($> 4.0$) [43]. For the remaining variables, the non-parametric Friedman test was used with the Wilcoxon signed rank test used as post hoc. The correlation coefficient (r) was calculated using the Wilcoxon Z-scores and the total number of observations (N) (i.e., $r = Z/\sqrt{N}$) and considered as 0.1 to $< 0.3$ (small), 0.3 to $< 0.5$ (moderate) and $\geq 0.5$ (large) [44]. The level of statistical significance was set at $p \leq 0.05$.

## Results

### Agility performance

There was a main significant effect of conditions ($F_{5,11}$ = 37.3; p < 0.001; $\eta p^2$ = 0.94). CAF+M condition elicited better performance than control (95%CI$_{diff}$: -0.91 to -0.50; p < 0.001; d = -2.47), NoS+M (95%CI$_{diff}$: -0.78 to -041; p < 0.001; d = -2.55), CAF+NoM (95%CI$_{diff}$: -0.50 to -0.16; p < 0.001; d = -1.37), PL+NoM (95%CI$_{diff}$: -0.89 to -0.45; p < 0.001; d = -2.79), and the PL+M (95%CI$_{diff}$: -0.72 to -0.28; p < 0.001; d = -1.88) conditions (Table 1).

### Ten s frequency speed of kicks (FSKT-10s)

There was a main significant effect of conditions ($Chi^2$ = 63.20; N = 16; df = 5; p < 0.001). CAF+M condition elicited better performance than control (r = 0.89; p < 0.001), NoS+M (r = 0.88; p < 0.001), CAF+NoM (r = 0.86; p = 0.001), PL+NoM (r = 0.89; p < 0.001), and PL +M (r = 0.88; p < 0.001; d = 3.64) conditions (Table 1).

### Multiple frequency speed of kicks (FSKT-mult)

There was a main significant effect of conditions ($Chi^2$ = 68.05; N = 16; df = 5; p < 0.001). CAF+M elicited better performance than control (r = 0.88; p < 0.001), NoS+M (r = 0.88; p < 0.001), CAF+NoM (r = 0.89; p < 0.001), PL+NoM (r = 0.88; p < 0.001) and PL+M (r = 0.88; p < 0.001) conditions (Table 1).

### Perceived exertion

**RPE_TSAT.** There was a main significant effect of conditions ($Chi^2$ = 39.03; N = 16; df = 5; p < 0.001). CAF+M condition elicited lower values than control (r = 0.86; p < 0.001),

**Table 1. Physical performance of taekwondo athletes during the taekwondo specific agility test (TSAT), 10s frequency speed of kick test (FSKT-10s) and the multiple version of FSKT (FSKT-mult) in the different conditions (n = 16).**

| | Control | | NoS+M | | CAF+NoM | | PL+NoM | | PL+M | | CAF+M | |
|---|---|---|---|---|---|---|---|---|---|---|---|---|
| | Mean (SD) | Med/ IQR | Mean (SD) | Med/ IQR | Mean (SD) | Med/ IQR | Mean (SD) | Med/ IQR | Mean (SD) | Med/ IQR | Mean (SD) | Med/ IQR |
| TSAT (s) | 5.67 (0.32) | - | 5.55 (0.21) | - | 5.29 (0.23)*$£ | - | 5.63 (0.23) | - | 5.46 (0.28)ᵠ | - | 4,96 (0.25) *$$£∂ | - |
| FSKT-10s (n) | 25.69 (0.87) | 26/1 | 27 (1.03)ᵃᵠ | 27/ 1.75 | 28.44 (1.03) *ᵇ£¥ | 28.5/1 | 25.44 (1.03) | 25/1 | 27.44 (1.46)ᵃᵠ | 27.5/2 | 32,13 (1.09) *$$£ ∂ | 32/2 |
| FSKT-mult (n) | 124.81 (1.56) | 125/ 2.75 | 127.38 (1.31)ᵃᵠ | 127.5/ 2.75 | 129.56 (1.09) *ᵇ£¥ | 129.5/ 1.75 | 125 (1.86) | 125/ 3.5 | 128 (2.13)ᵃᵠ | 128/ 3.5 | 141 (1.32) *$$£∂ | 141/2 |

*: Different from control at p < 0.001;

ᵃ: different from control at p < 0.05;

$: different from NoS+M at p < 0.001;

ᵇ: different from NoS+M at p < 0.05;

§: different from CAF+NoM at p < 0.001;

£: different from PL+NoM at p < 0.001;

ᵠ: different from PL+NoM at p < 0.05;

∂: different from PL+M at p < 0.001;

¥: different from PL+M at p < 0.05;

NoS+M: no supplement and music: CAF+NoM; caffeine with no music; PL+NoM; placebo with no music; PL+M; placebo and music; CAF+M: caffeine and music; s: second; n: number of techniques; Med/IQR: median/interquartile range.

NoS+M (r = 0.81; p < 0.001), CAF+NoM (r = 0.62; p = 0.013), PL+NoM (r = 0.84; p = 0.001) and PL+M (r = 0.75; p = 0.003) conditions (Table 2).

**RPE_FSKT-10s.** There was a main significant impact of conditions ($Chi^2$ = 44.68; N = 16; df = 5; p < 0.001). CAF+M condition elicited lower values than control (r = 0.89; p < 0.001), NoS+M (r = 0.88; p < 0.001), CAF+NoM (r = 0.89; p < 0.001), PL+NoM (r = 0.89; p < 0.001), and PL+M (r = 0.80; p = 0.001) conditions (Table 2).

**RPE_FSKT-mult.** There was a main significant conditions effect ($Chi^2$ = 58.66; N = 16; df = 5; p < 0.001). CAF+M elicited lower values than control (r = 0.89; p < 0.001), NoS+M

**Table 2. Perceived exertion (RPE), physical enjoyment (PACES), feeling scale (FS), and felt arousal scale (FAS) of taekwondo athletes in the different conditions (n = 16).**

| | | Control | | NoS+M | | CAF+NoM | | PL+NoM | | PL+M | | CAF+M | |
|---|---|---|---|---|---|---|---|---|---|---|---|---|---|
| | | Mean (SD) | Med/ IQR | Mean (SD) | Med/ IQR | Mean (SD) | Med/ IQR | Mean (SD) | Med/ IQR | Mean (SD) | Med/ IQR | Mean (SD) | Med/ IQR |
| RPE | TSAT | 5.25 (0.86) | 5/1 | 4 (0.89)[a] | 4/1.75 | 3.69(1.20) [a][c] | 4/1.75 | 4.5 (0.89) [a] | 4.5/1 | 3.94 (1.12) [a] | 4/2 | 2.38 (0.89) [*∂cd] | 2/1 |
| | FSKT-10s | 7.19 (0.66) | 7/1 | 5.88 (0.96) [a] | 6/2 | 6 (0.63) [a] | 6/0 | 6.06 (1.39) [a] | 6/2 | 5.81 (1.38) [a] | 5.5/2 | 3.31 (1.20) [*$φ$£] | 3.5/2 |
| | FSKT-mult | 9.56 (0.51) | 10/1 | 8.56 (0.51) [*c d] | 9/1 | 7.63 (0.96) [*bc£] | 8/1 | 9.38 (0.62) | 9/1 | 9.25 (0.68) | 9/1 | 6.38 (1.31) [*$$∂d] | 6.5/2.75 |
| PACES | TSAT | 61.69 (3.94) [bc] | 62/6.25 | 54.19 (4.56) | 54.5/ 5.75 | 70.69(4.16) [a$$£] | 71/5.5 | 55.88 (3.40) | 55/5.25 | 59 (5.54) [b] | 59.5/8 | 73.75 (4.81) [a$$£] | 74/7.5 |
| | FSKT-10s | 63.25(5.03) [cd] | 63/8.25 | 59.19(5.11) | 59.5/ 8.5 | 59.94(5.25) | 59.5/8.5 | 58.81 (3.47) | 60/5.5 | 57.13 (5.86) | 56.5/3.5 | 68.88 (4.15) [ab∂$£] | 69/6.5 |
| | FSKT-mult | 63.38(4.47) [$d] | 64.5/ 8.75 | 62.13(3.84) [$d] | 63/4.5 | 65.94(7.78) [$d] | 64/6.75 | 52.06 (4.07) | 51.5/6 | 57.5 (5.07) [c] | 58/8 | 67.88 (5.51) [b $£] | 68/7 |
| FS | TSAT | 0.13(0.96) | 0/1 | 1.13 (0.89) [ac] | 1/1.75 | 1.38(0.96) [a][c] | 1/1 | 0.13 (1.45) | 0/2 | 1.13 (0.72) [ac] | 1/1 | 3.38 (1.15) [*$φ$£] | 3.5/2 |
| | FSKT-10s | -1.06(1.06) | -1/2 | -0.94(1.57) | -0.5/2 | 0.81 (0.83) [*bc] | 1/1.75 | -0.38 (1.54) | -1/2.75 | -0.25 (1.73) | 0/3.5 | 2.06 (0.85) [*$∂cd] | 2/1.5 |
| | FSKT-mult | -4.13(1.26) | -4/1 | -0.38(1.41) [*] | 0/2.75 | -0.13 (1.20) [*c] | 0/2 | -1.5 (1.37) [*] | -1/1 | -0.38 (1.41) [*c] | -0.5/ 1.75 | 2 (1.03)[*$φ$£] | 2/1.75 |
| FAS | Warm-up | 1.56(0.73) | 1/1 | 3.5(0.89) [∂c] | 4/1 | 2.69 (0.79) [ac] | 2.5/1 | 2 (0.63) | 2/0 | 3.13 (0.81) [*c] | 3/1.75 | 5.13 (0.72) [*$φ$£] | 5/1 |
| | TSAT | 1.38(0.62) | 1/1 | 2.44(0.89) [ac] | 2/1 | 2.63 (0.89) [*c] | 2.5/1 | 1.31 (0.48) | 1/1 | 2.44 (1.03) [ac] | 2/1 | 4.06 (1.44) [*bc∂d] | 4/2 |
| | FSKT-10s | 1.38(0.50) | 1/1 | 1.56(0.63) | 1.5/1 | 3(0.82) [ab$] | 3/2 | 1.5 (0.73) | 1/1 | 2.63 (0.89) [*bc] | 2.5/1 | 4.88 (0.72) [*$φ$£] | 5/1 |
| | FSKT-mult | 1.19(0.40) | 1/0 | 1.5(0.52) | 1.5/1 | 2(0.73) [abcd] | 2/1.5 | 1.25 (0.45) | 1/0.75 | 1.44 (0.51) | 1/1 | 4.75 (0.77) [*$φ$£] | 5/1 |

*: Different from control at p < 0.001;

[a]: different from control at p < 0.05;

$: different from NoS+M at p < 0.001;

[b]: different from NoS+M at p < 0.05;

§: different from PL+NoM at p < 0,001;

[c]: different from PL+NoM at p < 0.05;

£: different from PL+M at p < 0.001;

[d]: different from PL+M at p < 0.05;

φ: different from CAF+Nom at p < 0.001;

∂: different from CAF+NoM at p < 0.05;

NoS+M: no supplement and music; CAF+NoM: caffeine with no music; PL+NoM: placebo with no music; PL+M: placebo and music; CAF+M: caffeine and music; Med/IQR: median/interquartile range.

(r = 0.86; p = 0.001), PL+NoM (r = 0.89; p < 0.001), CAF+NoM (r = 0.61; p = 0.016) and PL +M (r = 0.86; p = 0.001) conditions (Table 2).

## Felt arousal scale

**FAS_ post-warm-up.** There was a main significant effect of conditions ($Chi^2$ = 60.16; N = 16; df = 5; p < 0.001). CAF+M elicited greater performance than control (r = 0.89; p < 0.001), NoS+M (r = 0.84; p = 0.001), CAF+NoM (r = 0.89; p < 0.001), PL+NoM (r = 0.89; p < 0.001) and PL+M (r = 0.87; p = 0.001) conditions (Table 2).

**FAS_TSAT.** There was a main significant effect of conditions ($Chi^2$ = 45.34; N = 16; df = 5; p < 0.001). CAF+M resulted in higher values than control (r = 0.86; p = 0.001), NoS+M (r = 0.70; p = 0.005), PL+NoM (r = 0.78; p = 0.002), CAF+NoM (r = 0.61; p = 0.014), and PL +M (r = 0.69; p = 0.006) conditions (Table 2).

**FAS_FSKT-10s.** There was a main significant effect of conditions ($Chi^2$ = 58.19; N = 16; df = 5; p < 0.001). CAF+M induced higher values than control (r = 0.90; p < 0.001), NoS+M (r = 0.90; p < 0.001), CAF+NoM (r = 0.83; p = 0.001), PL+NoM (r = 0.89; p < 0.001) and the PL+M (r = 0.89; p < 0.001) conditions (Table 2).

**FAS_FSKT-mult.** There was a main significant effect of conditions ($Chi^2$ = 54.64; N = 16; df = 5; p < 0.001). CAF+M induced higher values than control (r = 0.89; p < 0.001), NoS+M (r = 0.90; p < 0.001), CAF+NoM (r = 0.89; p < 0.001), PL+NoM (r = 0.89; p < 0.001) and PL +M (r = 0.90; p < 0.001) conditions (Table 2).

## Feeling scale

**FS_TSAT.** There was a main significant effect of conditions ($Chi^2$ = 45.39; N = 16; df = 5; p < 0.001). CAF+M resulted in higher values than control (r = 0.86; p = 0.001), NoS+M (r = 0.83; p = 0.001), CAF+NoM (r = 0.84; p = 0.001), PL+NoM (r = 0.86; p = 0.001) and PL +M (r = 0.87; p = 0.001) conditions (Table 2).

**FS_FSKT-10s.** There was a main significant effect of conditions ($Chi^2$ = 34.38; N = 16; df = 5; p < 0.001). CAF+M elicited higher values than control (r = 0.89; p < 0.001), NoS+M (r = 0.83; p = 0.001), CAF+NoM (r = 0.75; p = 0.003), PL+NoM (r = 0.78; p = 0.002) and PL +M (r = 0.77; p = 0.002) conditions (Table 2).

**FS_FSKT-mult.** There was a main significant effect of conditions ($Chi^2$ = 57.06; N = 16; df = 5; p < 0.001). CAF+M elicited higher values than control (r = 0.89; p < 0.001), NoS+M (r = 0.80; p = 0.001), CAF+NoM (r = 0.83; p = 0.001), PL+NoM (r = 0.89; p < 0.001) and PL +M (r = 0.86; p = 0.001) conditions (Table 2).

## Physical activity enjoyment

**PACES_TSAT.** There was a main significant effect of conditions ($Chi^2$ = 57.40; N = 16; df = 5; p < 0.001). CAF+M induced higher values than control (r = 0.87; p = 0.001), NoS+M (r = 0.88; p < 0.001), PL+NoM (r = 0.88; p < 0.001) and PL+M(r = 0.88; p < 0.001) conditions (Table 2).

**PACES_FSKT-10s.** There was a main significant effect of conditions ($Chi^2$ = 31.26; N = 16; df = 5; p < 0.001). CAF+M elicited higher values than control (r = 0.71; p = 0.005), NoS+M (r = 0.86; p = 0.001), CAF+NoM (r = 0.82; p = 0.001), PL+NoM (r = 0.88; p < 0.001) and PL+M (r = 0.82; p = 0.001) conditions (Table 2).

**PACES_FSKT-mult.** There was a main significant effect of conditions ($Chi^2$ = 46.09; N = 16; df = 5; p < 0.001). CAF+M elicited higher values than NoS+M (r = 0.70; p = 0.005), PL +NoM (r = 0.88; p < 0.001), and PL+M (r = 0.84; p = 0.001) conditions (Table 2).

Full raw data are provided in S2 Table.

### Gastro-intestinal discomfort

Before CAF supplementation, three athlete reported moderate diarrhea symptom (i.e., 3 points) when conducting the CAF+NoM condition. After the test sessions, in the CAF+M condition, two athletes showed moderate headache symptom, four athletes showed moderate diarrhea symptom and three other athletes showed moderate abdominal pain. In the CAF +NoM condition, two athletes showed moderate headache symptom and five showed moderate diarrhea symptom.

## Discussion

The present study showed that consuming a low dose of CAF and listening to music during warm-up sessions improved physical performance in all taekwondo-specific tests compared to the other conditions. Furthermore, the CAF+M condition resulted in more positive psychological responses (i.e., higher PACES, FAS and FS scores) and lower RPE than the other conditions. The results of the present study supported our hypothesis.

Since studies about the acute combined effects of CAF and music is currently lacking, it seems difficult to compare our findings with those previously reported. However, regarding the effects of CAF and music in separate use, our results support previous studies [12, 22, 23, 31, 34, 45, 46]. In fact, 3 mg·kg⁻¹ of CAF resulted in the improvement of the number of throws during the special judo fitness test (SJFT) by 5% [45], jump height by 2.7% [31], number of techniques during the FSKT-10s by 4% and reduced the completion time during TSAT by 3.3% [8, 12]. This improvement could be explained by the CAF effect at both peripheral and central levels [27]. Specifically, at the peripheral level, CAF enhances calcium ion ($Ca^{2+}$) mobilization, and sodium/potassium ($Na^+/K^+$) pump activity, which may potentially enhance excitation-contraction coupling, motor unit recruitment, and synchronization [27, 47]. At the central level, CAF serves as an adenosine antagonist, improving neurotransmitter production and nervous system activation [48], which all allow CAF to be efficient for boosting muscle power production and movement velocity [49]. Regarding music effects, the findings from the present study confirm the previously reported benefits of listening to preferred music on exercise performance [22, 23, 34]. In fact, it has been suggested that listening to preferred music during warm-up raised anticipatory response to exercise, extending effort and muscle force development [22]. Considerably, the related benefits of self-selected warm-up music is its efficiency in boosting maximal effort [23] particularly in short-term anaerobic exercise [46]. For instance, Meglic et al. [22] found that 3 min of standardized cycling warm-up at 50 watts while listening to preferred music increased mean power in a subsequent high-intensity repeated exercise (i.e., 3 x 15 s Wingate). As well, catecholamine release's increase following listening to preferred music, influenced muscle activation and metabolic responses during subsequent exercise [50] possibly altering blood flow and lactate clearance [17]. Consequently, the association between the CAF effectiveness on neural activity and glycolytic metabolism and music efficacy on metabolites clearance could generate a favorable condition to perform at high-intensity tasks.

Considering the psychological responses, PACES, FS and FAS scores were higher during CAF+M compared to other conditions. These results could be comparable to those reported in previous studies [1, 51] where 6 mg·kg⁻¹ of CAF ingested 60 min prior to testing enhanced vigor, tension and subjective vitality profiles. Moreover, 3 mg·kg⁻¹ of CAF 60 min prior to the same tests used in the present study, Ouergui et al. [12] reported mood and physical symptoms increase in taekwondo athletes without affecting subjective vitality profile and FS. Therefore, CAF supplementation may induce an optimal emotional state to promote better physical performance and prepare the athlete to deal with fatiguing situations [1]. The improvement of

these parameters following CAF ingestion may be attributed to the enhancement of brain activation [1] and the increased central motivation to exercise [52]. The benefits from listening to music during warm-up were allocated to affective states' improvement during intense exercise [21, 53], through bringing back good memories [54] and improve athletes' emotional state and self-efficacy [21, 53]. When athletes listened to their preferred music during warm-up, performance gains were accompanied by both motivation and arousal increase [22, 34]. The influence of music on athletes' level of activation is closely dependent on its acoustical properties [54], highlighting the suitability of fast upbeat music to fast power type activities [21]. In the present study, the tempos of selected-music were higher than 120 bpm which can be considered as stimulating [16]. Consequently, physical enjoyment, feelings, and perceived arousal in the present study were improved with the central effects of both music and CAF.

It is well known that the ergogenic potential of CAF is established by its direct action on the central nervous system, leading to reduce subjective exhaustion [2]. However, CAF suppressor effect on RPE was not confirmed in combat sports [8]. In this consideration, Ouergui et al. [12] reported that the same dose of CAF (i.e., 3 mg/kg of body mass) improved physical performances without affecting athletes' perceived exertion. Moreover, using higher dose of CAF (i.e., 6 mg/kg) 60 min before the 30s Wingate test, Jodra et al. [51] did not report a significant effect on RPE in male boxers. However, using similar procedures (i.e., 6 mg/kg of CAF 60 min prior to the Wingate test), Dominguez et al. [1] showed reduced RPE values in males resistance-trained participants. Although the inconsistent findings might indicate moderating effects of population characteristics and training background [8], the recorded effect in the present study could suggest that replacing an extra amount of CAF with musical stimulus had similar effects to high doses without inducing significant side effects. Noticeably, RPE has been reported to be the most constant factor which is influenced by music during exercise [17]. The significant influence of music during exercise on RPE is generally attributed to dissociation from discomfort and exertion [17]. However, the effects of warm-up music remained unclear. The analgesic effects of CAF might support the extension of the ergogenic effects of warm-up music recorded in the present study. Since adenosine genesis primarily regulated by energy deprivation, CAF antagonism of adenosine signaling can prevail this regulation, resulting in fatigue reduction and increased enjoyment even when energy levels are low [7]. Such effect of CAF on RPE has been suggested to be mediated by its impact on cerebral oxygenation [1].

The effect of CAF was reported to be minimized in highly trained subjects [55] and music effects may be neutralized by significant interceptive cues of physical tenderness associated with the exercise when performing at high-intensity [19, 54]. In the present study, combining both ergogenic aids enhanced performances in elite taekwondo athletes, which could be attractive findings for athletes from high competitive levels. The synergetic mechanisms explaining the improvement of physical performances and psychological responses resulting from the combination between CAF and music are unclear. However, the action mechanisms of music and CAF could explain these results. The positive action of CAF on psychological aspects could improve the athlete's behaviors to exercise, which probably improve alertness and reaction time [2]. Furthermore, it was revealed that the effects of CAF on cognition and brain activation are greater with low doses of CAF than with moderate and high doses [56]. Thus, 3 mg. kg$^{-1}$ of body mass ingested in the present study could be effective to block the inhibitor effects of adenosine [26, 56], counteracting the negative effect of adenosine on central fatigue. Regarding music, it is well known that it can result in increased motivation, activity enjoyment, self-confidence, feelings of power, and regulated arousal [17–20]. These effects may be of high levels when using preferred music, as music selection may have a key role in deciding whether music acts as an ergogenic aid or not [23]. Consequently, the reported outcomes might be the result of synergetic effects at both the central and peripheral levels, since CAF

and music commonly serve to reduce perceived exertion [7, 17, 52], increase muscular activation [47, 57] and improve affective state [1, 5, 6, 22, 34, 51, 53].

While the present study resulted in novel approach able to enhance taekwondo athletes' performance, some limitations should be acknowledged. The findings from this study are specific to taekwondo male athletes and should be taken with caution for other sports modalities as well for female athletes. Moreover, the tests used are specific and could mimic the competition demands, but they did not inform us about the effects of this combination in a real competition setting to investigate the technical-tactical behaviors.

## Conclusions

The present study showed that a low dose of CAF combined with listening to preferred music during warm-up improved physical-specific performances and psychological state in elite taekwondo athletes and reduced their perceived exertion better than their use separately. Moreover, for highly trained athletes, combining CAF and warm-up music serves to extend their effects during repeated high-intensity specific exercises as indicated using the FSKT-mult. Furthermore, the improvement recorded through the combination between low dose of CAF consumption and the use of musical stimulus during warm-up was not accompanied by significant side effects, indicating such a strategy to be safe and efficient.

## Supporting information

**S1 Table. Consolidated Standards of Reporting Trials (CONSORT) checklist of the study.**
(DOCX)

**S2 Table. Full raw data.**
(XLSX)

## Acknowledgments

The authors thanks the athletes participated in this study.

## Author Contributions

**Conceptualization:** Slaheddine Delleli, Ibrahim Ouergui, Hamdi Messaoudi, Christopher Garrett Ballmann, Luca Paolo Ardigò, Hamdi Chtourou.

**Data curation:** Slaheddine Delleli, Ibrahim Ouergui, Hamdi Messaoudi, Christopher Garrett Ballmann, Luca Paolo Ardigò, Hamdi Chtourou.

**Formal analysis:** Slaheddine Delleli, Ibrahim Ouergui, Hamdi Messaoudi, Christopher Garrett Ballmann, Luca Paolo Ardigò, Hamdi Chtourou.

**Investigation:** Slaheddine Delleli, Ibrahim Ouergui, Hamdi Messaoudi, Christopher Garrett Ballmann, Luca Paolo Ardigò, Hamdi Chtourou.

**Methodology:** Slaheddine Delleli, Ibrahim Ouergui, Hamdi Messaoudi, Christopher Garrett Ballmann, Luca Paolo Ardigò, Hamdi Chtourou.

**Project administration:** Slaheddine Delleli, Ibrahim Ouergui, Hamdi Messaoudi, Christopher Garrett Ballmann, Luca Paolo Ardigò, Hamdi Chtourou.

**Resources:** Slaheddine Delleli, Ibrahim Ouergui, Hamdi Messaoudi, Christopher Garrett Ballmann, Luca Paolo Ardigò, Hamdi Chtourou.

**Software:** Slaheddine Delleli, Ibrahim Ouergui, Hamdi Messaoudi, Christopher Garrett Ballmann, Luca Paolo Ardigò, Hamdi Chtourou.

**Supervision:** Slaheddine Delleli, Ibrahim Ouergui, Hamdi Messaoudi, Christopher Garrett Ballmann, Luca Paolo Ardigò, Hamdi Chtourou.

**Validation:** Slaheddine Delleli, Ibrahim Ouergui, Hamdi Messaoudi, Christopher Garrett Ballmann, Luca Paolo Ardigò, Hamdi Chtourou.

**Visualization:** Slaheddine Delleli, Ibrahim Ouergui, Hamdi Messaoudi, Christopher Garrett Ballmann, Luca Paolo Ardigò, Hamdi Chtourou.

**Writing – original draft:** Slaheddine Delleli, Ibrahim Ouergui, Hamdi Messaoudi, Christopher Garrett Ballmann, Luca Paolo Ardigò, Hamdi Chtourou.

**Writing – review & editing:** Slaheddine Delleli, Ibrahim Ouergui, Hamdi Messaoudi, Christopher Garrett Ballmann, Luca Paolo Ardigò, Hamdi Chtourou.

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
