## [Decision Letter · Decision Letter 0]

25 Aug 2023

PONE-D-23-20580Effects of Caffeine Consumption Combined with Listening to Music during Warm-up on Taekwondo Physical Performance, Perceived Exertion and Psychological AspectsPLOS ONE

Dear Dr. Ardigò,

Thank you for submitting your manuscript to PLOS ONE. After careful consideration, we feel that it has merit but does not fully meet PLOS ONE’s publication criteria as it currently stands. Therefore, we invite you to submit a revised version of the manuscript that addresses the points raised during the review process.

We look forward to receiving your revised manuscript.

Kind regards,

Goran Kuvačić, PhD

Academic Editor

PLOS ONE

https://www.mdpi.com/2304-6767/10/8/147

In your revision ensure you cite all your sources (including your own works), and quote or rephrase any duplicated text outside the methods section. Further consideration is dependent on these concerns being addressed.

Reviewers' comments:

Reviewer's Responses to Questions

**Comments to the Author**

1. Is the manuscript technically sound, and do the data support the conclusions?

Reviewer #1: Yes

Reviewer #2: Yes

2. Has the statistical analysis been performed appropriately and rigorously? 

Reviewer #1: Yes

Reviewer #2: Yes

3. Have the authors made all data underlying the findings in their manuscript fully available?

Reviewer #1: Yes

Reviewer #2: Yes

4. Is the manuscript presented in an intelligible fashion and written in standard English?

Reviewer #1: Yes

Reviewer #2: Yes

5. Review Comments to the Author

Reviewer #1: The authors aimed analyze the acute responses of combining a (low) dose of caffeine (CAF) with warm-up music on physical performance of taekwondo, perceived exertion,

and psychological responses during taekwondo-specific tasks in elite athletes. They organized a

double-blinded, randomized, placebo-controlled crossover study design, asking male taekwondo athletes to perform the taekwondo-specific agility test, 10 s frequency speed of kick test (FSKT-10s) and the multiple versions of FSKT (FSKT-mult) varying the conditions of caffeine and music.

The better performances were using caffeine and music in comparison to other conditions for

TSAT, FSKT-10s, FSKT-mult, RPE, FAS and FS and PACES post_FSKT-10s. Moreover, CAF+M resulted in better responses than other conditions for PACES post TSAT with the exception of CAF+NoM. Likewise, CAF+M condition induced better physical enjoyment than other conditions post FSKT-mult (p<0.05). Combining low dose of CAF with music during warm-up was an effective strategy that induced greater effects than their isolated use during taekwondo specific tasks. The study is very interesting, and I have just some minor points. The written is clear and the study is well organized. The methods and results are well written, and the discussion present the rationale and benchmark with the literature accordingly.

My unique major point is in the introduction where the hypothesis is not well supported from the literature or the rationale from why additional gains were expected using music and caffeine in comparison to other conditions. Which mechanisms possibly are operating to give this response. And after discussing that in detail in the discussion.

I suggest including a checklist to report the study accordingly. Specifically, CONSORT 2010 statement: extension to randomized crossover trials:

https://www.equator-network.org/reporting-guidelines/consort-2010-statement-extension-to-randomised-crossover-trials/

Minor points

Abstract

Perceived exertion is no abbreviated in the first point.

Line 78 – that3

Line 132 – PACES was already abbreviated (consider review all abbreviations).

Line 440 – consider using effective instead of efficient.

The tables are well designed though I’d avoid vertical lines in tables.

Reviewer #2: Line 23: Start the abstract with ‘what is known’ and ‘what is not known’ to justify the study

Line 44: can you add some examples of ergogenic effects?

Line 51: can you add some examples of different performance aspects?

Line 90: please explain how the subjects were recruited and add the criteria for inclusion and exclusion

Line 136: how long takes the washout?

Line 157: the reference is not in journal style

Line 158: one decimal seems enough

Line 338: can you confirm your hypothesis?

Line 377: space after reference

6. PLOS authors have the option to publish the peer review history of their article (what does this mean?). If published, this will include your full peer review and any attached files.

Reviewer #1: No

Reviewer #2: No

---

## [Author Response · Author response to Decision Letter 0]

11 Sep 2023

REVIEW

Plos One

PONE-D-23-20580

Dear editors and reviewers:

Enclosed is the revised version of the manuscript PONE-D-23-20580, entitled “Effects of caffeine consumption combined with listening to music during warm-up on taekwondo physical performance, perceived exertion and psychological aspects". We would like to express our gratitude to both editor and reviewers for their thorough review of our work and for their constructive and helpful comments, which have greatly improved the quality of our manuscript. We have made the necessary changes to the manuscript, indicated in red, and have also attached a document that addresses the reviewers’ queries point by point. We hope that these revisions have improved the manuscript and that it is now suitable for publication in your journal. Additionally, we are ready to make any further changes that may be necessary for further improvement.

Reviewer 1: 

The authors aimed analyze the acute responses of combining a (low) dose of caffeine (CAF) with warm-up music on physical performance of taekwondo, perceived exertion,

and psychological responses during taekwondo-specific tasks in elite athletes. They organized a double-blinded, randomized, placebo-controlled crossover study design, asking male taekwondo athletes to perform the taekwondo-specific agility test, 10 s frequency speed of kick test (FSKT-10s) and the multiple versions of FSKT (FSKT-mult) varying the conditions of caffeine and music.

The better performances were using caffeine and music in comparison to other conditions for

TSAT, FSKT-10s, FSKT-mult, RPE, FAS and FS and PACES post_FSKT-10s. Moreover, CAF+M resulted in better responses than other conditions for PACES post TSAT with the exception of CAF+NoM. Likewise, CAF+M condition induced better physical enjoyment than other conditions post FSKT-mult (p<0.05). Combining low dose of CAF with music during warm-up was an effective strategy that induced greater effects than their isolated use during taekwondo specific tasks. The study is very interesting, and I have just some minor points. The written is clear and the study is well organized. The methods and results are well written, and the discussion presents the rationale and benchmark with the literature accordingly.

Authors’ response

We thank the expert reviewer for his/her comment.

Comment 1

My unique major point is in the introduction where the hypothesis is not well supported from the literature or the rationale from why additional gains were expected using music and caffeine in comparison to other conditions. Which mechanisms possibly are operating to give this response. And after discussing that in detail in the discussion.

Authors’ response 

We thank the expert reviewer for his/her comment. Some changes were added in the text as follows: “While music is a psychological ergogenic aid that serves to improve exercise behaviors and affective states [1], CAF is a pharmaceutical substance which showed its effectiveness to enhance neurological and physiological responses to exercise [2]. Given that performance is an integration of physical capacities, physiological responses and psychological behaviors [3],….”.

Comment 2

I suggest including a checklist to report the study accordingly. Specifically, CONSORT 2010 statement: extension to randomized crossover trials:

https://www.equator-network.org/reporting-guidelines/consort-2010-statement-extension-to-randomised-crossover-trials/

Authors’ response 

Thank you for your suggestion. A The study was reported following the CONSORT 2010 statement checklist for randomized crossover trials and a CONSORT checklist was added as a supplementary file.

Minor points

Abstract

Comment 3

Perceived exertion is no abbreviated in the first point.

Authors’ response 

Thank you. The abbreviation of Perceived exertion was added at its first appearance. Please find changes in the text. 

Comment 4

Line 78 – that3

Authors’ response 

The space was added, thank you.

Comment 5

Line 132 – PACES was already abbreviated (consider review all abbreviations).

Authors’ response 

Abbreviation was removed

Comment 6

Line 440 – consider using effective instead of efficient.

Authors’ response 

The word was changed accordingly. 

Comment 7

The tables are well designed though I’d avoid vertical lines in tables.

Authors’ response 

Thank you. Vertical lines were removed

We appreciate the expert reviewer's comments and we hope that the work met the requirements to be accepted for publication.

Reviewer 2: 

Comment 1

Line 23: Start the abstract with ‘what is known’ and ‘what is not known’ to justify the study

Authors’ response 

We thank the expert reviewer for his/her comment. The following sentences were added as follows: “The effects of caffeine (CAF) and music have been well documented when used separately, but their combined effects are not yet studied. Thus, the present study assessed the acute effects of combining a low dose of caffeine (CAF) with listening to music during warm-up on taekwondo physical performance, perceived exertion (RPE), and psychological responses during taekwondo-specific tasks in male elite athletes.

Comment 2

Line 44: can you add some examples of ergogenic effects?

Authors’ response 

We thank the reviewer for his/her comment. Examples of ergogenic effects are already reported in the sentences from line 46as follows: “Ergogenic potential of CAF intake has been widely established for many exercises types, including muscular endurance/ strength, anaerobic power, and aerobic endurance [4]. The ergogenic potential of CAF is generally explained by its capacity to block the adenosine receptors A1, A2A, and A2B , due to their similarity in term of chemical structures [5]. In addition, CAF supplementation has been shown to promote a positive mood [6, 7], to increase alertness and reduces the feeling of fatigue [8].”

Comment 3

Line 51: can you add some examples of different performance aspects?

Authors’ response 

We thank the expert reviewer for the comment. Performance aspects were described as follow: “In combat sports, CAF has been reported to improve different performance aspects involving isometric strength, anaerobic power, reaction time, and anaerobic metabolism”.

Comment 4

Line 90: please explain how the subjects were recruited and add the criteria for inclusion and exclusion

Authors’ response 

The following information were added as follows: “16 male taekwondo athletes from the national team (Mean ± SD; age: 18.25 ± 0.75 years; body mass: 60.92 ±8.96 kg; height: 182 ±6.84 cm) volunteered to participate in the present study. Athletes were recruited following a convenience sampling based on the following eligibility criteria: a) being an elite athlete with at least 8 years of taekwondo experience; b) do not suffer from any restrictions to sports practice or hearing impairments; c) have at least 17 years old, and d) being non-smoker”

Comment 5

Line 136: how long takes the washout?

Authors’ response 

Thank you for the comment. Seven days washout period was used. This was already mentioned in the manuscript as follows: “The sessions were separated by an interval of seven days to allow sufficient recovery between sessions and to ensure CAF washout [9]”. Additionally, this information was added in the flow diagram.

Comment 6

Line 157: the reference is not in journal style

Authors’ response 

The references’ style was rechecked and was made in journal style when there was mistake.

Comment 7

Line 158: one decimal seems enough

Authors’ response 

One decimal was used. Thank you

Comment 8

Line 338: can you confirm your hypothesis?

Authors’ response 

Hypothesis was confirmed and the following sentences were added at the beginning of the discussion as follows. “The results of the present study supported our hypothesis”

Comment 9

Line 377: space after reference

Authors’ response 

The space was added. Thank you.

We thank the expert reviewer for his/her comments and hope that the article met the requirements to be accepted for publication.

---

## [Editor Report · Decision Letter 1]

21 Sep 2023

Effects of caffeine consumption combined with listening to music during warm-up on taekwondo physical performance, perceived exertion and psychological aspects

PONE-D-23-20580R1

Dear Dr. Ardigò,

We’re pleased to inform you that your manuscript has been judged scientifically suitable for publication and will be formally accepted for publication once it meets all outstanding technical requirements.

Kind regards,

Goran Kuvačić, PhD

Academic Editor

PLOS ONE
---

## [Editor Report · Acceptance letter]

12 Oct 2023

PONE-D-23-20580R1 

Effects of caffeine consumption combined with listening to music during warm-up on taekwondo physical performance, perceived exertion and psychological aspects 

Dear Dr. Ardigò:

I'm pleased to inform you that your manuscript has been deemed suitable for publication in PLOS ONE. Congratulations! Your manuscript is now with our production department. 

Kind regards, 

on behalf of

Dr. Goran Kuvačić 

Academic Editor

PLOS ONE